# Polysubstituted High-Entropy [LaNd](Cr_0.2_Mn_0.2_Fe_0.2_Co_0.2_Ni_0.2_)O_3_ Perovskites: Correlation of the Electrical and Magnetic Properties

**DOI:** 10.3390/nano11041014

**Published:** 2021-04-15

**Authors:** Vladimir E. Zhivulin, Evgeniy A. Trofimov, Svetlana A. Gudkova, Igor Yu. Pashkeev, Alexander Yu. Punda, Maksim Gavrilyak, Olga V. Zaitseva, Sergey V. Taskaev, Fedor V. Podgornov, Moustafa A. Darwish, Munirah A. Almessiere, Yassine Slimani, Abdulhadi Baykal, Sergei V. Trukhanov, Alex V. Trukhanov, Denis A. Vinnik

**Affiliations:** 1Laboratory of Single Crystal Growth, South Ural State University, 454080 Chelyabinsk, Russia; zhivulinve@susu.ru (V.E.Z.); tea7510@gmail.com (E.A.T.); svetlanagudkova@yandex.ru (S.A.G.); pashkeevii@susu.ru (I.Y.P.); pundaai@susu.ru (A.Y.P.); gavrilyak.maksim@yandex.ru (M.G.); nikonovaolga90@gmail.com (O.V.Z.); s.v.taskaev@gmail.com (S.V.T.); fedorpod@yahoo.de (F.V.P.); denisvinnik@gmail.com (D.A.V.); 2Department of Technology of Electronics Materials, National University of Science and Technology MISiS, Leninsky Prospekt 4, 119049 Moscow, Russia; mostafa_ph@science.tanta.edu.eg (M.A.D.); sv_truhanov@mail.ru (S.V.T.); 3Physics Department, Faculty of Science, Tanta University, Al-Geish st., Tanta 31527, Egypt; 4Department of Biophysics, Institute for Research and Medical Consultations (IRMC), Imam Abdulrahman Bin Faisal University, P.O. Box 1982, Dammam 31441, Saudi Arabia; malmessiere@iau.edu.sa (M.A.A.); yaslimani@iau.edu.sa (Y.S.); 5Department of Nanomedicine Research, Institute for Research and Medical Consultations (IRMC), Imam Abdulrahman Bin Faisal University, P.O. Box 1982, Dammam 31441, Saudi Arabia; abaykal@iau.edu.sa; 6Laboratory of Magnetic Films Physics, Scientific-Practical Materials Research Centre of National Academy of Sciences of Belarus, 220072 Minsk, Belarus

**Keywords:** high-entropy oxides, high-entropy perovskite, multiple substitution, configuration entropy

## Abstract

La-, Nd- and La/Nd-based polysubstituted high-entropy oxides (HEOs) were produced by solid-state reactions. Composition of the B-site was fixed for all samples (Cr_0.2_Mn_0.2_Fe_0.2_Co_0.2_Ni_0.2_) with varying of A-site cation (La, Nd and La_0.5_Nd_0.5_). Nominal chemical composition of the HEOs correlates well with initial calculated stoichiometry. All produced samples are single phase with perovskite-like structure. Average particle size is critically dependent on chemical composition. Minimal average particle size (~400 nm) was observed for the La-based sample and maximal average particle size (5.8 μm) was observed for the Nd-based sample. The values of the configurational entropy of mixing for each sample were calculated. Electrical properties were investigated in the wide range of temperatures (150–450 K) and frequencies (10^−1^–10^7^ Hz). Results are discussed in terms of the variable range hopping and the small polaron hopping mechanisms. Magnetic properties were analyzed from the temperature and field dependences of the specific magnetization. The frustrated state of the spin subsystem was observed, and it can be a result of the increasing entropy state. From the Zero-Field-Cooling and Field-Cooling regimes (ZFC-FC) curves, we determine the <S> average and S_max_ maximum size of a ferromagnetic nanocluster in a paramagnetic matrix. The <S> average size of a ferromagnetic cluster is ~100 nm (La-CMFCNO) and ~60 nm (LN-CMFCNO). The S_max_ maximum size is ~210 nm (La-CMFCNO) and ~205 nm (LN-CMFCNO). For Nd-CMFCNO, spin glass state (ferromagnetic cluster lower than 30 nm) was observed due to f-d exchange at low temperatures.

## 1. Introduction

Functional materials with controlled properties based on many composite oxides are of great interest due to fundamental and practical aspects [1,2,3]. Transition metal ions-based complex oxides are the most important class of magnetic materials. They are widely discussed among researchers and find various practical applications as magnets with high coercivity and remnant magnetization [4,5], as magnetic memory materials [6], for catalysis [7], biomedical applications [8,9,10] and as electromagnetic absorbers of high-frequency radiation [11,12,13,14]. Mn-based complex oxides with perovskite structure (or manganites) attract great attention due to strong correlation of the chemical composition, cation ordering, crystal structure peculiarities and unique magnetic, electrical and magnetotransport properties [15,16,17,18]. Substituted manganites with general formula ABO_3_ (where A^3+^—rare earth element such as Ln = La, Pr, Nd or substituted element; B—Mn) have a perovskite-like structure. Both A and B cations form a spatial cubic lattice in an ideal perovskite, with both partial lattices shifted relative to each other by half of the spatial diagonal of the basic cube. The space group (SG) of an ideal undistorted perovskite is Pm3m [16]. Chemical substitution of the A-site cations (Ln^3+^) can change the valence state of the Mn^3+^ [16] or induce deviation from oxygen stoichiometry [19]. As a result, it leads to changes in magnetic ordering and electrical properties.

Interest in manganites is due to the strong correlation of chemical composition, structural, magnetic and electrical parameters, and the ability to influence physical properties by changing the chemical composition. The chemical composition of manganites and their local atomic structure are two main parameters that determine the intensity of indirect exchange interactions, which is manifested by changing <Mn-O> bond lengths and <Mn-O-Mn> angles. Bond lengths and bond angles determine magnetic and electrical parameters. One of the main approaches to improving the properties of manganites is chemical substitution of Mn3+ ions by diamagnetic and paramagnetic ions.

The amounts of dopants and their oxidation state strongly correlate with the value of the system entropy of perovskites. The use of dia- or para-magnetic substituents with a higher/lower ionic radius and oxidation states, as a rule, increases the disorder (or entropy) in the system, and therefore causes a change in the magnetic and electrical characteristics. Recently, the prospects of the development and application of high-entropy systems were discussed in many papers [20,21]. Most often, we are talking about high-entropy metal alloys; however, the study of non-metallic high-entropy systems has significantly advanced in recent years. An example is the work devoted to the study of the synthesis and the properties of high-entropy oxides, or HEOs [22,23,24,25,26,27,28,29,30]. Earlier researchers initially studied simple oxide compounds based on divalent metals (Mg, Co, Ni, Cu and Zn, for example, in [22,27,28]) or trivalent metals [30,31]. However, their interest was quickly attracted by HEO compounds with a more complex crystal structure than systems formed only by oxides of tri- or bi-valent metals [32,33,34,35,36]. Phases of high-entropy oxides with a spinel-like crystal structure were obtained in [37,38]. The authors of [39,40,41,42] synthesized high-entropy phases with a crystal structure similar to perovskite. The main problem is to achieve a single-phase composition. High configurational entropy of mixing makes it possible to stabilize a multicomponent crystal lattice at concentrations of components close to equimolar.

The aim of this work is to study the possibility of synthesizing a high-entropy La/Nd manganite with multiple substitution in B-sites (Cr_0.2_Mn_0.2_Fe_0.2_Co_0.2_Ni_0.2_) with a perovskite-like structure and investigation of the correlation between the A-site composition (La, Nd and La_0.5_Nd_0.5_), magnetic, electrical properties and entropy configuration. Presumably, a high value of the entropy of mixing should stabilize multicomponent La(Cr_0.2_Mn_0.2_Fe_0.2_Co_0.2_Ni_0.2_)O_3_ (or La-CMFCNO), Nd(Cr_0.2_Mn_0.2_Fe_0.2_Co_0.2_Ni_0.2_)O_3_ (or Nd-CMFCNO) and [La_0.5_Nd_0.5_](Cr_0.2_Mn_0.2_Fe_0.2_Co_0.2_Ni_0.2_)O_3_ (or LN-CMFCNO) solid solutions with a perovskite-like structure. The main idea of the choice of research objects is to establish a correlation between the ionic radius of the A-cation (La, Nd and the La_0.5_Nd_0.5_ intermediate composition), the state of configurational entropy and the features of the magnetic and transport properties. Thus, the La^3+^ ion is a diamagnetic ion and is characterized by the maximum ionic radius and leads to distortion of the lattice, which in turn leads to an increase in exchange interactions. The Nd^3+^ ion is a paramagnetic ion, and at low temperatures (below 50 K), magnetic ordering effects can be observed (the neodymium sublattice can be ordered antiparallel to the manganese sublattice, which leads to compensation effects). The aim of the study of the intermediate compound (La_0.5_Nd_0.5_ perovskite) is to evaluate the simultaneous influence of the diamagnetic lanthanum ion and the paramagnetic neodymium ion in the structure of high-entropy perovskite.

## 2. Materials and Methods

La-CMFCNO, Nd-CMFCNO and LN-CMFCNO ceramic samples, where CMFCNO in B-site was fixed (Cr_0.2_Mn_0.2_Fe_0.2_Co_0.2_Ni_0.2_) with varying of the A-site cation (La, Nd and La_0.5_Nd_0.5_), were produced by the solid-state reactions method. The initial components for the preparation of samples were powders of lanthanum (III) (La_2_O_3_) and neodymium (III) (Nd_2_O_3_) oxides, iron (III) oxide (Fe_2_O_3_), chromium (III) oxide (Cr_2_O_3_), manganese (III) oxide (Mn_2_O_3_), cobalt (II) oxide (CoO) and nickel (II) oxide (NiO). All the chemicals used were of analytical grade (99.999%). All compounds were synthesized according to the following equations:
(1)$$\text{La-CMFCNO:}\;0.5{\mathrm{La}}_{2}O_{3}+\left\{0.1{\mathrm{Cr}}_{2}O_{3}+0.1{\mathrm{Mn}}_{2}O_{3}+0.1{\mathrm{Fe}}_{2}O_{3}+0.2\mathrm{CoO}+0.2\mathrm{NiO}\right\}+0.1O_{2}\to \left[\mathrm{La}\right]\{{\mathrm{Cr}}_{0.2}{\;\mathrm{Mn}}_{0.2}{\mathrm{Fe}}_{0.2}{\mathrm{Co}}_{0.2}{\mathrm{Ni}}_{0.2}\}O_{3}$$
(2)$$\text{Nd-CMFCNO:}\;0.5{\mathrm{Nd}}_{2}O_{3}+\left\{0.1{\mathrm{Cr}}_{2}O_{3}+0.1{\mathrm{Mn}}_{2}O_{3}+0.1{\mathrm{Fe}}_{2}O_{3}+0.2\mathrm{CoO}+0.2\mathrm{NiO}\right\}+0.1O_{2}\left[\mathrm{Nd}\right]\{{\mathrm{Cr}}_{0.2}{\;\mathrm{Mn}}_{0.2}{\mathrm{Fe}}_{0.2}{\mathrm{Co}}_{0.2}{\mathrm{Ni}}_{0.2}\}O_{3}$$
(3)$$\text{LN-CMFCNO:}\;\left[0.25{\mathrm{La}}_{2}O_{3}+0.25{\mathrm{Nd}}_{2}O_{3}\right]+\left\{0.1{\mathrm{Cr}}_{2}O_{3}+0.1{\mathrm{Mn}}_{2}O_{3}+0.1{\mathrm{Fe}}_{2}O_{3}+0.2\mathrm{CoO}+0.2\mathrm{NiO}\right\}+0.1O_{2}\to \left[{\mathrm{La}}_{0.5}{\mathrm{Nd}}_{0.5}\right]\{{\mathrm{Cr}}_{0.2}{\;\mathrm{Mn}}_{0.2}{\mathrm{Fe}}_{0.2}{\mathrm{Co}}_{0.2}{\mathrm{Ni}}_{0.2}\}O_{3}$$

The composition of the material was selected based on the need to maximize the configurational entropy of mixing of one of the cationic sublattices. It was assumed that such a high-entropy B-site sublattice was formed by Cr, Mn, Fe, Co and Ni, and the A-site sublattice was formed by lanthanum and neodymium cations. The configurational entropy of La-CMFCNO, Nd-CMFCNO and LN-CMFCNO within each sublattice of ceramic samples can be obtained using the following equation:ΔS^conf^_sublatt_ = −R Σ x_i_·ln x_i_(4)
where R—gas constant (8.314 J·K^−1^⋅mol^−1^) and x_i_—atomic fraction of sublattice component.

The maximum value of ΔS^conf^_sublatt_ for a given number of components is achieved if the concentrations of the components are equimolar. For a sublattice consisting of 5 components (B-site sublattice), the maximum value (the value 1.609R J/mole) is achieved at concentrations of Cr, Mn, Fe, Co and Ni ions equal to 0.2. For a sublattice consisting of 2 components (A-site sublattice in LN-CMFCNO), the maximum value (the value 0.693R J/mole) is achieved at La and Nd ion concentrations equal to 0.5. It is these concentrations that reflect Formulas (1)–(3).

The initial components (oxides) were mixed in the appropriate stoichiometric proportions and ground during 3 h in a ball mill. Good long-term mixing is essential for optimal homogenization and single-phase synthesis. Mixed powders were compacted in pellets (diameter and height were 1 cm) at 3 tons/cm^2^. In the course of the experiment, weighed portions of the resulting mixture were heated and kept in an oven at 1300 °C in air for 10 h. The heating rate was 400 °C/h. The cooling rate was 100 °C/h. The phase composition and crystal structure were investigated utilizing X-ray powder diffraction (XRD) in Cu-Kα radiation. The measurements were performed on a diffractometer Rigaku Ultima IV in the angular range from 10° to 80° with the speed of 2°/min. The average chemical composition was controlled using the scanning electron microscope (SEM) Jeol JSM7001F with energy dispersive spectrometer Oxford INCA X-max 80 by energy-dispersive X-ray spectroscopy (EDX). Magnetic properties were investigated by VersaLab Quantum Design Physical Properties Measurement System (PPMS) in the wide temperature and magnetic field ranges. Field dependencies were measured at fixed temperatures (50 and 300 K) in the range of ±3T. Electrical properties were measured on a Beta Single-Unit Dielectric Analyzer in the frequency range 10^−1^–10^7^ Hz and temperature range 123–473 K. Variants of the sublattice model are usually used for the thermodynamic description of complex oxide systems. According to this model, the configuration of the system is defined by a conjunction of two or more pure sublattices formed by particles of different types. Here, the total configurational entropy of mixing is limited by the entropy of mixing separately in each of the sublattices, and also determined by the ratio of the total number of particles belonging to different sublattices.

## 3. Results and Discussion

### 3.1. Chemical Composition and Structural Features

Nominal chemical composition, crystal structure and grain morphology of the as-prepared La-CMFCNO, Nd-CMFCNO and LN-CMFCNO ceramic samples were investigated using EDX, XRD and SEM, respectively. Table 1 shows the elemental composition (at. %) of the resulting ceramics according to EDX spectroscopy data. The gross formula was calculated from the elemental analysis data.

It can be seen from the table that the gross formula correlates well with the calculated data. The deviation from the initial stoichiometry (calculated formula) for all samples was no higher than 5%. Nominal chemical composition (real formula) of the La-CMFCNO corresponded to the La_1.04_(Cr_0.22_Mn_0.21_Fe_0.18_Co_0.17_N_i0.19_)O_3_, nominal chemical composition of the Nd-CMFCNO corresponded to the Nd_1.01_(Cr_0.21_Mn_0.21_Fe_0.21_Co_0.19_Ni_0.18_)O_3_ and nominal chemical composition of the LN-CMFCNO corresponded to the [La_0.53_Nd_0.48_](Cr_0.21_Mn_0.20_Fe_0.19_Co_0.19_Ni_0.2_)O_3_. Thus, the provided gross chemical formula correlates well with stoichiometry.

Crystal structure of the La-CMFCNO, Nd-CMFCNO and LN-CMFCNO ceramic samples was tested by XRD analysis using Cu-Kα radiation at room temperature. Figure 1 shows the XRD patterns for all high-entropy oxides (HEOs). The main structural parameters obtained from XRD data after refinements (lattice parameters, relevance factors and atomic coordination) for all samples are presented in Table 2.

It was observed that all samples exhibit single-phase compounds. There was no secondary phase observed. La-CMFCNO possesses hexagonal symmetry and space group (SG) R-3c or #167 (code No. 98-004-9739 in ICSD). At the same time, it was observed that the other two HEO samples (Nd-CMFCNO and LN-CMFCNO) can be described by the orthorhombic symmetry and SG: Pnma or #62 (code No. 98-003-7439 ICSD). For all XRD data, the Rietveld refinements were done for calculation of the lattice parameters and to determine the atomic coordination. XRD patterns for all investigated samples have demonstrated appropriate values of the relevance factors (χ^2^, R_p_, R_WP_ and R_exp_). The volume of the unit cell for La-CMFCNO (350.231 Å^3^) is greater than for Nd-CMFCNO (227.019 Å^3^) and LN-CMFCNO (230.017 Å^3^). Differences in volume of the unit cell can be explained by the different ionic radii of the A-site cations. The ionic radii of A-site cations are La^3+^ = 1.30 Å and Nd^3+^ = 1.25 Å, respectively.

The microstructure of the investigated sample obtained with a scanning electron microscope (second electron mode, at ×2500 magnification) is shown in Figure 2. The set of SEM images was used for statistical analysis of particle size using the software SmartSEM. The size measurements were carried out with the assumption that the shape of the particles is round. The proportion of particles’ area (*P*) was calculated by the equation:(5)$$P=\frac{\pi }{4}\frac{d_{i}^{2}n_{i}}{S},$$
where *d_i_* is the size (equivalent disk diameter) of each measured particle, *S* is the full area of all particles under consideration and *n_i_* is the number of particles with a selected size (for example 0–500 nm, 510–1000 nm and so on).

All particle diameter distributions were divided into 7–20 fractions with a 500 nm step. The distributions were plotted according to the measurements and are presented in Figure 3.

Figure 2a shows the microstructure of La-CMFCNO sample. The average particle size of this sample is approximately 400 nm. At the same time, the width of the particles’ size distribution (black graph in Figure 3) is the narrowest, since the maximum particle size does not exceed 3.5 μm (the distribution width at half the height is 1.4 µm). The microstructure of Nd-CMFCNO is shown in Figure 2b. The microstructure of Nd-based perovskite differs significantly from lanthanum-based perovskite. The Nd-CMFCNO sample is characterized by a high density and absence of pores compared to the La-CMFCNO. The average particle size for the Nd-based sample was 5.8 μm, which is 4 times larger than that for La-based perovskite. It should be noted that the particle size varies very widely from 1.5 to 10 μm, as shown by the distribution (red graph) in Figure 3. Figure 2c shows the microstructure of LN-CMFCNO perovskite. It was found that the sample consists mostly of particles with an average size of about 4 μm. Larger particles are surrounded by agglomerates of finer particles with average size around 1 μm. The distribution unambiguously demonstrates bimodal behavior of particle size. The average particle size of the fine phase is about 300–400 nm. Agglomerates of a fine phase are localized at the boundaries of large particles, thereby filling voids and pores, which makes it possible to increase the density of the material.

### 3.2. Configuration Entropy

Using standard equations, we can discuss the calculation of the configurational entropy (∆Sconf) of samples (in J per 1 mole of metal ions). The calculation was carried out based on the idea that metal ions in A- and B-sites, together with the surrounding oxygen atoms (in an amount sufficient to compensate for the positive charge of metal ions), form separate sublattices. The configurational entropy of La-CMFCNO, Nd-CMFCNO and LN-CMFCNO ceramic samples for this model can be obtained using Equation (5).

Using the data in Table 1, we calculated the real values of ΔS^conf^_sublatt_ for B-sublattice La-CMFCNO (1.601R J/mole), Nd-CMFCNO (1.608R) and LN-CMFCNO (1.609R), as well as ΔS^conf^_sublatt_ for A-sublattice LN-CMFCNO (0.692R). As shown, these values differ very little from the ideal 1.609R and 0.693R J/mole.

Calculation of the average entropy of mixing per 1 mole of cations uses the equation:(6)$${\Delta S}^{\mathrm{conf}}\mathrm{cation}=-\frac{\sum_{j}(x_{i}^{sub\_1}\ln x_{i}^{sub\_1})+\sum_{j}(x_{i}^{\begin{array}{l} sub\_2 \end{array}}\ln x_{i}^{sub\_2})}{2}*R$$
where $x_{i}^{sub\_n}$—mole fraction of *i* metal cations in cation sublattice n. For La-CMFCNO and Nd-CMFCNO samples, the average values are 0.800R and 0.804R J/mole respectively, and for LN-CMFCNO, the sample average value is around 1.151R J/mole.

### 3.3. Electrical Properties

To reveal the mechanisms of charge carrier transport in the investigated materials, the conductivity spectra were measured with the impedance spectrometer Novocontrol Beta System in the temperature range 123–453 K. Within this temperature interval, the investigated material is paramagnetic. The frequency of the probing voltage signal was in the range 0.1 Hz–10 MHz and the amplitude was 5 mV. The log-log plots of the real parts of the electric conductivity spectra ($\sigma^{\prime }$) for different temperatures are shown in Figure 4. As one can see, $\sigma^{\prime }$ follows the Jonscher power law:(7)$$\sigma^{\prime }=\sigma_{DC}+\sum A_{j}f^{s_{j}}$$
where $\sigma_{DC}$ is the direct current (DC) electric conductivity, *f* is the frequency of the probing voltage and $A_{j}$ and $s_{j}$ are parameters describing alternating current (AC) electric conductivity.

As follows from (7), $\sigma_{DC}$ is the vertical asymptote to $\sigma^{\prime }$ curve at low frequency limit.

Usually, the $\sigma^{\prime }$ grows with the increase of applied voltage frequency. In other words, they follow the Jonscher theory. Such functional dependence could be observed in the “low” temperature range (T < 333 K). At high temperatures (T > 333 K), one can see the opposite functional dependence, which is in contradiction to the universal Jonscher law. Usually, it is explained in the framework of the classical Drude model. This model well-describes frequency dependences of the $\sigma^{\prime }$ for La-CMFCNO.

In the frequency range 10–10^5^ Hz, electrical conductivity was constant for all temperatures. At higher frequencies (10^5^–10^7^ Hz), temperature’s strong impact on the conductivity was observed. $\sigma^{\prime }$ increased at frequencies higher than 10^5^ Hz at temperatures of 123–323 K, and at temperatures of 373–473 K, we observed the opposite behavior (Figure 4a).

For the Nd-CMFCNO sample (Figure 4b), the conductivity spectra do not have practical frequency dispersion and depend only on temperature. For all investigated samples, an increase of the temperature leads to an increase of the electrical conductivity. This is a standard situation for the semimetals (perovskites). For the LN-CMFCNO sample, at frequencies higher than 10^5^ Hz, a rapid decrease of $\sigma^{\prime }$ for all temperatures was observed (Figure 4c), which can be explained by the features of the charge relaxation at higher frequencies.

However, in this work, we will completely concentrate on the investigation of the DC conductivity. The studies of the AC conductivity will be the subject of another paper. The temperature dependences of $\sigma_{DC}$ for La-CMFCNO, Nd-CMFCNO and LN-CMFCNO ceramic samples are plotted in Figure 5. This graph grows with the increase of temperature and finally reaches saturation at T≈455 K (for La-CMFCNO), T≈375 K (for Nd-CMFCNO) and T≈310 K (for LN-CMFCNO). Hence, one can conclude that these compounds have a semiconducting type of conductivity.

In further discussions, we will consider only La-CMFCNO and Nd-CMFCNO components, because they should have non-trivial electric transport mechanisms. Depending on temperature, they have two conduction mechanisms: the variable range hopping (VRH) conduction mechanism (low-temperature range) and the small polaron hopping (SPH) conduction mechanism (high-temperature range).

The SPH conduction mechanism is observed at temperatures below half of the Debye temperature ($T<\frac{T_{D}}{2}$). This range is characterized by the strong electron–optical phonons interactions. The half Debye temperature could be found from $\ln\left(\frac{1}{\sigma_{DC}T}\right)vs\;\frac{1}{T}$ plot (see Figure 6 for La-CMFCNO and Nd-CMFCNO), as the temperature at which the second derivative changes the sign. For the investigated compound, the Debye temperature could be estimated as $T_{D}\approx 860\;K$ (La-CMFCNO) and $T_{D}\approx 640\;K$ (Nd-CMFCNO). The optical phonon frequency is estimated via Relation (8):(8)$$\omega_{ph}=\frac{k_{B}T_{D}}{h}$$
where $k_{B}$ and h are the Boltzmann and Planck constants, respectively.

From (8), it follows that the La-CMFCNO and Nd-CMFCNO have the phonon frequencies $\omega_{ph}\approx 1.72\times 10^{13}$ Hz and $\omega_{ph}\approx 1.37\times 10^{13}$ Hz, respectively.

In the temperature range $\frac{T_{D}}{4}<T<\frac{T_{D}}{2}$, the dominant conduction mechanism is the variable range hopping, and the temperature dependence of the DC conductivity obeys the Mott-VRH (Mott- Variable Range Hopping) law [43]:(9)$$\sigma_{DC}=\sigma_{0}e^{-{\left(\frac{T_{0}}{T}\right)}^{\frac{1}{4}}}$$
where $\sigma_{0}$—pre-exponential parameter, and $T_{0}$—a parameter depending on the density of states in Fermi level.

By plotting $\ln\left(\sigma_{DC}\right)$ as a function of $T^{-\frac{1}{4}}$ (Figure 7), one can observe the linear part of the curve for the temperature interval $\frac{T_{D}}{4}<T<\frac{T_{D}}{2}$, which corresponds to the VRH conduction mechanism for La-CMFCNO (Figure 7a). At the same time, for Nd-CMFCNO, this mechanism dominates for the whole temperature range of investigation (Figure 7b).

The SPH electric conductivity ($\sigma_{DC}$) depends on temperature and activation energy ($E_{A}$), as:(10)$$\sigma_{DC}=\frac{\sigma_{0}}{T}e^{-\frac{E_{A}}{k_{B}\;T}}$$
where $\sigma_{0}$ is a parameter.

Therefore, the activation energies could be derived from the $\ln\left(\sigma_{DC}T\right)vs\;\frac{1}{k_{B}\;T}$ plot (see Figure 8) by linear fitting of the curve in three different temperature ranges ($T>\frac{T_{D}}{4}$, $\frac{\;T_{D}}{4}<T<\frac{T_{D}}{2}$,$\;T<\frac{T_{D}}{2}$). The results of fitting are summarized in Table 3. The activation energy for Nd-CMFCNO is around 128 meV.

### 3.4. Magnetic Properties

The spin dynamics of the investigated compositions were studied in the so-called ZFC (zero-field-cooling) and FC (field cooling) modes. These data are shown in Figure 9.

In a weak field, the spins are very sensitive to the prehistory of measurements, and thus frustrated magnetic states can be detected [44]. The studied compositions are solid solutions with complex substitution in the B-sublattice of the ABO_3_ perovskite structure. Each B-cation separately forms an antiferromagnetic compound with both the La^3+^ lanthanum cation and the Nd^3+^ neodymium cation. The intensity of the B-O^2−^-B (B = Cr^3+^, Mn^3+/4+^, Fe^3+^, Co^3+/4+^, Ni^2+^) indirect super-exchange interactions is determined by the <r_A_> average radius of the A-sublattice of the perovskite structure. The higher the <r_A_> average radius, the more intensive the exchange. For example, the LaMnO_3_ lanthanum orthomanganite is the A-type, weak ferromagnet, with an ordering temperature of ~141 K [45], where the vanishing ferromagnetic moment is due to the asymmetric Dzyaloshinsky-Moriya exchange [46,47]. The highest temperature of magnetic ordering is possessed by the LaFeO_3_ lanthanum orthoferrite compound, which is the G-type antiferromagnet [48], for which the T_N_ Néel temperature is ~743 K [49]. For the Nd^3+^ neodymium compounds, the T_mo_ magnetic ordering temperature is much lower. So, the NdMnO_3_ neodymium orthomanganite is an antiferromagnet with a T_N_ Néel temperature equal to ~87 K [50,51], while for the NdFeO_3_ neodymium orthoferrite, it is ~688 K [49].

From the Figure 9a–c, a set of critical temperatures can be determined, and a number of conclusions can be drawn. The ordinate axis is not very informative in this case. Firstly, a significant splitting of the ZFC and FC curves is observed, which indicates a frustrated state of the spin subsystem. The presence of a complex substitution in the B-sublattice of the ABO_3_ perovskite structure decreases the T_mo_ magnetic ordering temperature of the studied solid solutions determined by the minimum of the FC curve derivative (see inset in Figure 9). Thus, in the case of the La^3+^ lanthanum cation, the observed T_mo_ magnetic ordering temperature is closest to the T_N_ Néel point for the LaMnO_3_ orthomanganite and it amounts to ~131 K. For this composition, one can also determine the so-called T_f_ freezing and T_div_ divergence temperatures [52]. According to the Bean–Livingston relationship [53], the T_f_ freezing temperature and the T_div_ divergence temperature determine the <S> average and S_max_ maximum size of a ferromagnetic cluster in a paramagnetic matrix. In the case of La^3+^ lanthanum cation, these temperatures are ~113 and ~227 K, respectively. According to our calculations, the <S> average size of a ferromagnetic cluster is ~100 nm, and the S_max_ maximum size reaches ~210 nm. Thus, it can be argued that there are ferromagnetic nanoclusters in the studied lanthanum composition. In addition, a noticeable temperature hysteresis is observed in the range of 100–200 K, which most likely indicates spin relaxation of a nonergodic system rather than a first-order phase transition due to its extension [54].

The intensity of the B-O^2−^-B indirect super-exchange interactions, as is well-known, is determined by the <B-O> average bond length and the <B-O-B> average bond angle [55]. The <r_A_> average radius of the A-sublattice of perovskite structure affects the values of both the <B-O> average bond length and the <B-O-B> average bond angle [56]. A decrease in the <r_A_> average radius of the A-sublattice increases the distortions of the unit cell and weakens the intensity of the indirect super-exchange interactions, which leads to a decrease in the <B-O-B> average bond angle and an increase in the <B-O> average bond length [57]. For the Nd^3+^ neodymium composition, the T_mo_ magnetic ordering temperature is ~53 K. Thus, conditions arise for frustration of the B-O^2−^-B indirect super-exchange interactions and the formation of a spin glass state [58]. However, in the case of Nd^3+^ neodymium cation, the peak on the ZFC curve is absent (see Figure 9b), which is associated with the effect of f-d exchange at low temperatures [59]. The frustration of the spins of the B cations is masked by the f-d exchange. The spin sublattice of the Nd^3+^ neodymium cations is ordered antiparallel to the spin sublattice of the B cations due to the negative f-d exchange [60]. At the T_comp_ compensation temperature of ~168 K, zero moment is observed, the temperature hysteresis is retained and the divergence temperature is ~191 K.

For the La^3+^-Nd^3+^ lanthanum-neodymium solid solution, the T_mo_ magnetic ordering temperature is ~65 K (see Figure 9c). The peak of the ZFC curve is fixed at ~93 K. An interesting feature is the presence of two compensation temperatures at ~64 and ~145 K. This fact can be explained by the formation of two microregions with an enriched and depleted content of the Nd^3+^ neodymium cations in the A-sublattice of perovskite structure. Regions enriched with the Nd^3+^ neodymium cations undergo spin disordering at higher temperatures. Temperature hysteresis is also observed. The divergence temperature increases up to ~218 K. According to our calculations, the <S> average size of a ferromagnetic cluster is ~60 nm, and the S_max_ maximum size reaches ~205 nm.

Figure 10a–c show the field magnetization loops of the studied solid solutions at different temperatures.

At room temperature, all the compositions are paramagnets, which can be seen from the linear dependence of the magnetization. At low temperatures, a spontaneous moment and magnetic hysteresis are observed. Magnetization is not saturated in fields up to 3 T. The maximum, Ms, spontaneous magnetization of ~15.2 emu/g is observed for the composition based on the La^3+^ lanthanum cation. The minimum, M_s_, spontaneous magnetization of ~0.6 emu/g is recorded for the mixed La^3+^-Nd^3+^ lanthanum-neodymium composition, although for the purely Nd^3+^ neodymium composition, the M_s_ spontaneous magnetization of ~0.7 emu/g is also very low. This is explained by the weakening of the intensity of the B-O^2−^-B indirect super-exchange interactions and the negative contribution of the Nd^3+^ neodymium sublattice in the magnetic field.

It seems interesting to generalize and trace the evolution of the magnetic parameters of the studied compositions depending on the <r_A_> average radius of the A-sublattice of the perovskite structure. This dependence is shown in Figure 11.

Three critical temperatures, such as the T_f_ freezing, T_div_ divergence and T_mo_ magnetic ordering, increase with an increase in the <r_A_> average ionic radius, and only one T_comp_ compensation temperature decreases under the same conditions. Moreover, the T_f_ freezing and T_div_ divergences temperatures increase with an increasing rate equal to ~648.9 and ~442.2 K/Å respectively, while the T_mo_ magnetic ordering temperature increases with an increasing rate of ~863.3 K/Å. According to our calculations, the <S> average and S_max_ maximum sizes of a ferromagnetic cluster vary within ranges of 60–100 and 205–210 nm, respectively. The T_comp_ compensation temperature decreases with an increase in the <r_A_> average ionic radius, with a decreasing rate of ~1.8 × 10^3^ K/Å. The M_s_ spontaneous magnetization increases with an increase in the <r_A_> average ionic radius, with an increasing rate of ~160.9 emu·g^−1^·Å^−1^. Thus, we can see a significant effect of the chemical composition and average ionic radius of the A- and B-sublattices of perovskite structure on the formation of the magnetic phase state and critical magnetic parameters of the studied solid solutions. The obtained results can be of interest from a practical point of view as materials for micro-electromechanical system (MEMS) magnetic sensing applications near room temperatures [61,62].

## 4. Conclusions

La-, Nd- and La/Nd-based polysubstituted [LaNd](Cr_0.2_Mn_0.2_Fe_0.2_Co_0.2_Ni_0.2_)O_3_ HEOs with perovskite-like structure were produced by solid-state reactions. Composition of the B-site was fixed for all samples (Cr_0.2_Mn_0.2_Fe_0.2_Co_0.2_Ni_0.2_), with varying of A-site cations (La, Nd and La_0.5_Nd_0.5_). From EDX data, it was established that the nominal chemical composition of the La(Cr_0.2_Mn_0.2_Fe_0.2_Co_0.2_Ni_0.2_)O_3_ (or La-CMFCNO), Nd(Cr_0.2_Mn_0.2_Fe_0.2_Co_0.2_Ni_0.2_)O_3_ (or Nd-CMFCNO) and [La_0.5_Nd_0.5_](Cr_0.2_Mn_0.2_Fe_0.2_Co_0.2_Ni_0.2_)O_3_ (or LN-CMFCNO) correlates well with initial calculated stoichiometry. The deviation from the initial stoichiometry (calculated formula) for all samples was no higher than 5%. Based on XRD data, all produced samples are single phase and can be described by the SG: R-3c (La-CMFCNO) and SG: Pnma (Nd-CMFCNO and LN-CMFCNO). Differences in volume of the unit cell can be explained by the different ionic radii of the A-site cations and local distortions under chemical substitution. It was observed that the average particle size critically depended on chemical composition of the investigated HEOs. Minimal average particle size (~400 nm) was observed for La-CMFCNO and maximal average particle size (5.8 μm) was observed for Nd-CMFCNO. LN-CMFCNO demonstrated bimodal behavior of particle size (with average sizes of about 4 μm and 300–400 nm). For La-CMFCNO and Nd-CMFCNO samples, the average value of the configurational entropy is around 0.805R J/mole, and for the LN-CMFCNO sample, the average value is around 1.151R J/mole. Electrical properties (conductivity) were investigated in the wide range of temperatures (150–450 K) and frequencies (10^−1^–10^7^ Hz). In the frequency range 10–10^5^ Hz, electrical conductivity was constant for all temperatures. At higher frequencies (10^5^–10^7^ Hz), it was observed that temperature had a strong impact on the conductivity. Results were discussed in terms of the variable range hopping and the small polaron hopping conduction mechanisms. Magnetic properties were analyzed from the temperature (ZFC-FC modes) and field dependences of the specific magnetization. In a weak field, the spins are very sensitive to the prehistory of measurements, and thus frustrated magnetic states can be detected. From the ZFC-FC curves, critical temperatures for the HEOs were determined: T_mo_—magnetic ordering temperature, T_comp_—compensation temperature, T_f_—freezing temperature and T_div_—divergence temperature. T_mo_ correlates well with ionic radii of the A-site cation (<r_A_>) and is equal to ~131.1 K (La-CMFCNO), ~53.4 K (Nd-CMFCNO) and ~64.9 K (LN-CMFCNO). In the case of the La-based sample, T_f_ and T_div_ are ~113 and ~227 K, respectively. A decrease in the <r_A_> average radius of the A-sublattice increases the distortions of the unit cell and weakens the intensity of the indirect super-exchange interactions, which leads to a decrease in the <B-O-B> average bond angle and an increase in the <B-O> average bond length. For the Nd-based sample, T_comp_ is ~168 K and T_div_ is ~191 K. In the case of the LaNd-based sample, T_div_ is ~218 K. Additionally, for this composition, two values of the T_comp_ at ~64 and ~ 145 K were observed. This fact can be explained by the formation of two microregions with an enriched and depleted content of the Nd^3+^ neodymium cations in the A-sublattice of perovskite structure. From the T_f_ freezing temperature and the T_div_ divergence temperature we determined the <S> average and S_max_ maximum size of a ferromagnetic nanocluster in a paramagnetic matrix. According to our calculations, the <S> average sizes of a ferromagnetic cluster were: ~100 nm (La-CMFCNO) and ~60 nm (LN-CMFCNO). The S_max_ maximum size reached ~210 nm (La-CMFCNO) and ~205 nm (LN-CMFCNO). For Nd-CMFCNO, spin glass state (ferromagnetic cluster lower than 30 nm) was observed due to f-d exchange at low temperatures. Thus, we can see a significant effect of the chemical composition and average ionic radius of the A- and B-sublattices of perovskite structure on the formation of the magnetic phase state and critical magnetic parameters of the studied HEOs.

## Figures and Tables

**Figure 1 nanomaterials-11-01014-f001:**
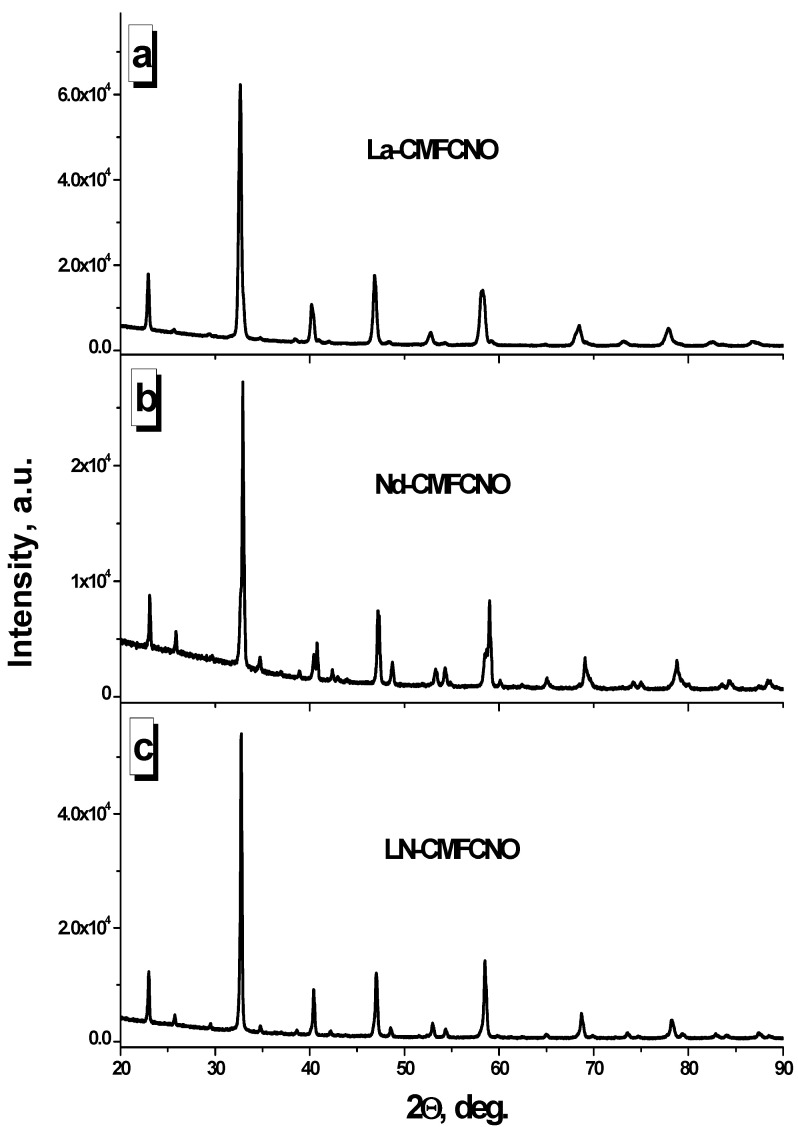
X-ray diffraction (XRD) patterns of the La-CMFCNO (**a**), Nd-CMFCNO (**b**) and LN-CMFCNO (**c**) ceramic samples.

**Figure 2 nanomaterials-11-01014-f002:**
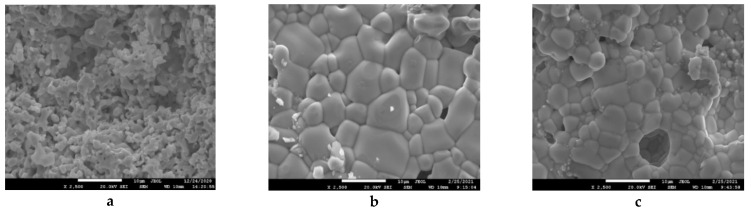
Scanning electron microscopy (SEM) images of the La-CMFCNO (**a**), Nd-CMFCNO (**b**) and LN-CMFCNO (**c**) ceramic samples.

**Figure 3 nanomaterials-11-01014-f003:**
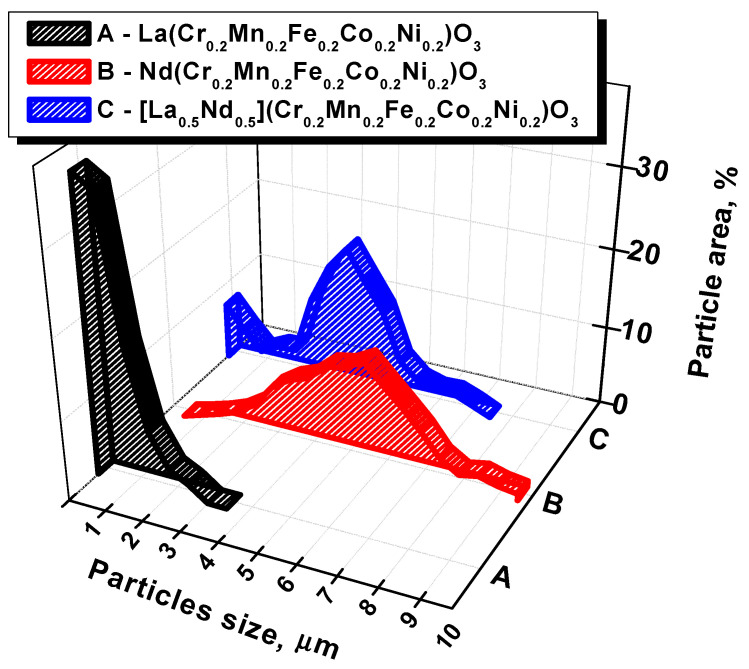
Particles’ size distribution of La-CMFCNO, Nd-CMFCNO and LN-CMFCNO ceramic samples.

**Figure 4 nanomaterials-11-01014-f004:**
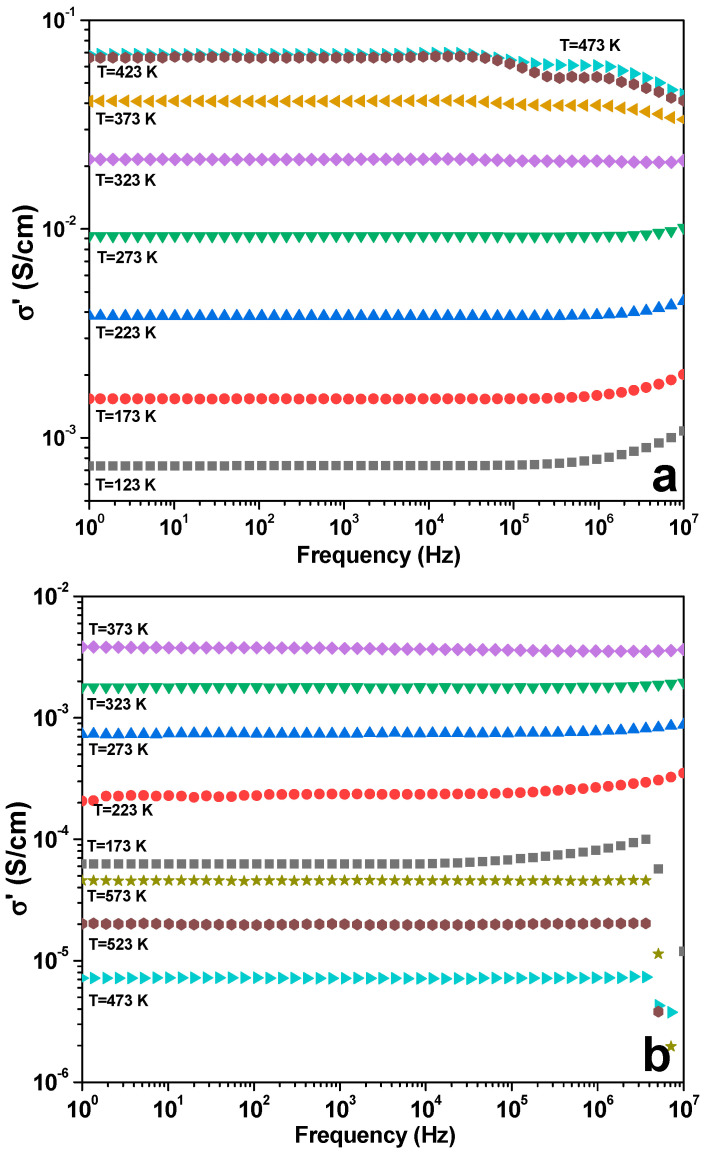
Spectra of the real part of conductivity at different temperatures for the La-CMFCNO (**a**), Nd-CMFCNO (**b**) and LN-CMFCNO (**c**) ceramic samples.

**Figure 5 nanomaterials-11-01014-f005:**
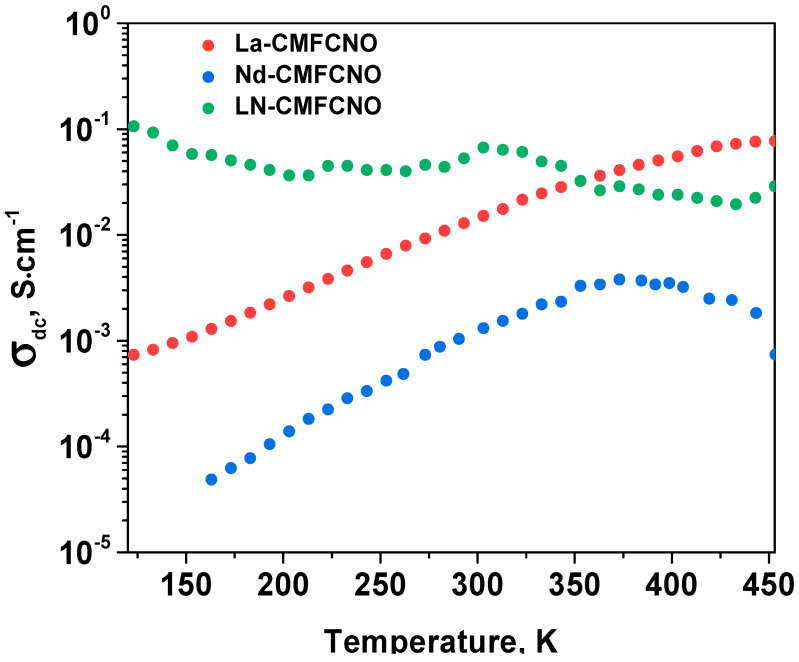
Temperature dependences of the σ_DC_ for the La-CMFCNO, Nd-CMFCNO and LN-CMFCNO ceramic samples.

**Figure 6 nanomaterials-11-01014-f006:**
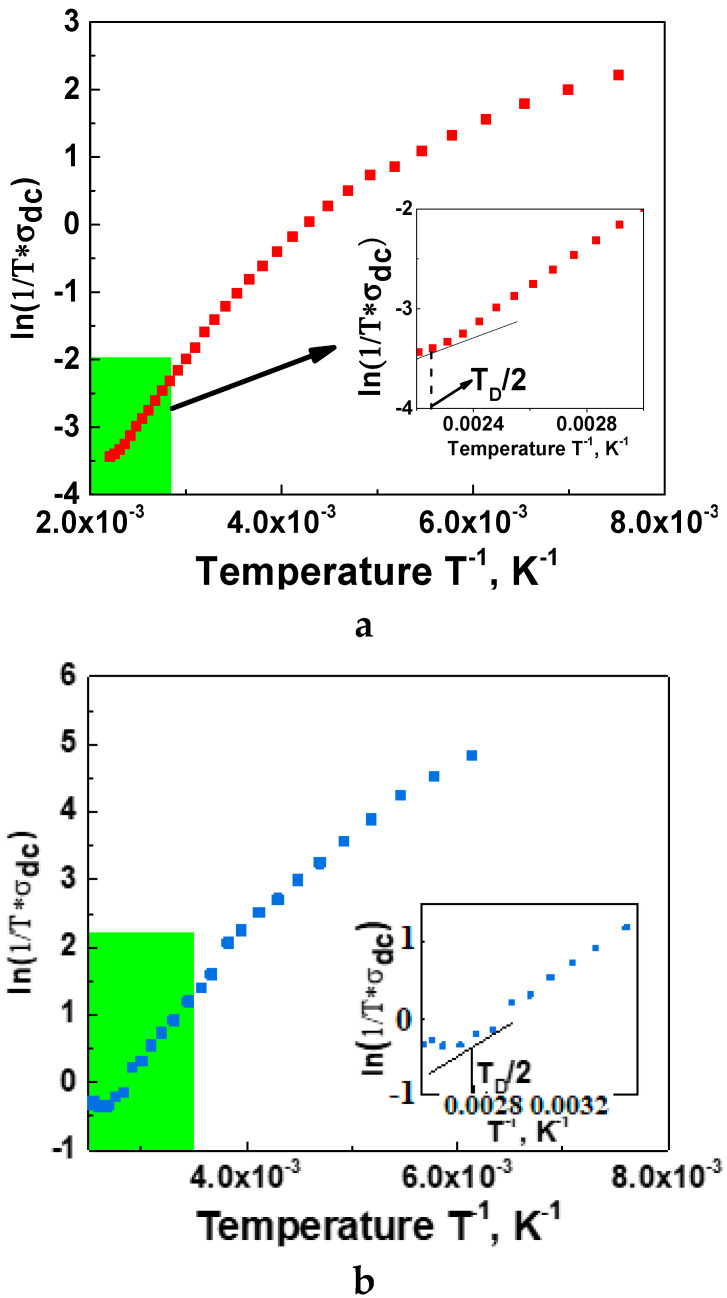
Dependence of ln(1/σ_DCT_) on 1/T. Estimation of Debye temperature for La-CMFCNO (**a**) and Nd-CMFCNO (**b**).

**Figure 7 nanomaterials-11-01014-f007:**
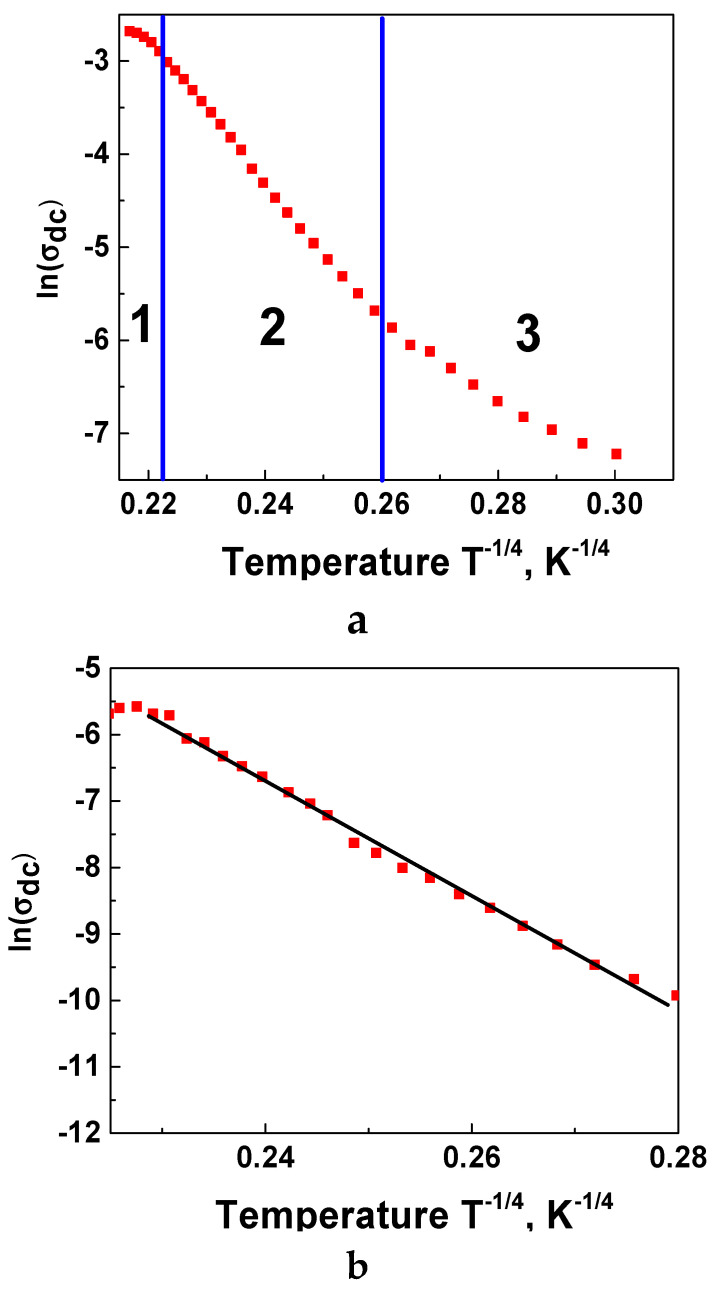
Dependence of ln(σ_DC_) on T^–1/4^ for La-CMFCNO (**a**) and Nd-CMFCNO (**b**).

**Figure 8 nanomaterials-11-01014-f008:**
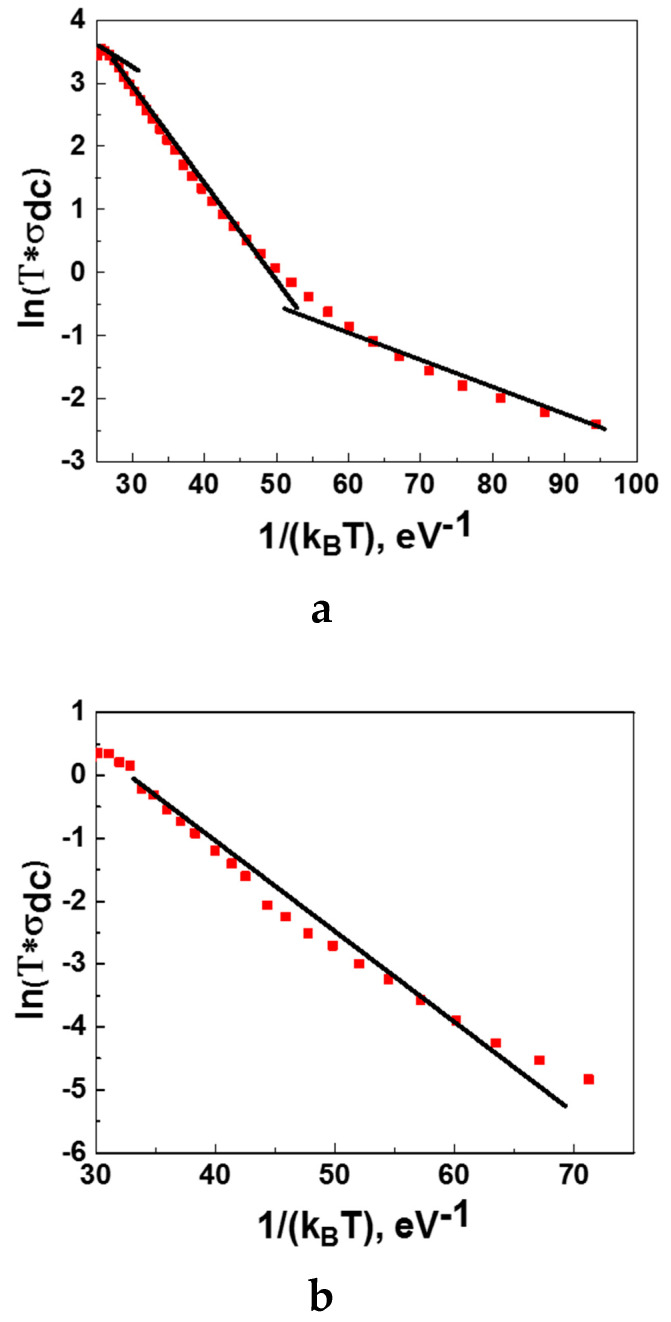
Dependence of ln(σ_DCT_) on $\frac{1}{k_{B}\;T}$ for La-CMFCNO (**a**) and Nd-CMFCNO (**b**).

**Figure 9 nanomaterials-11-01014-f009:**
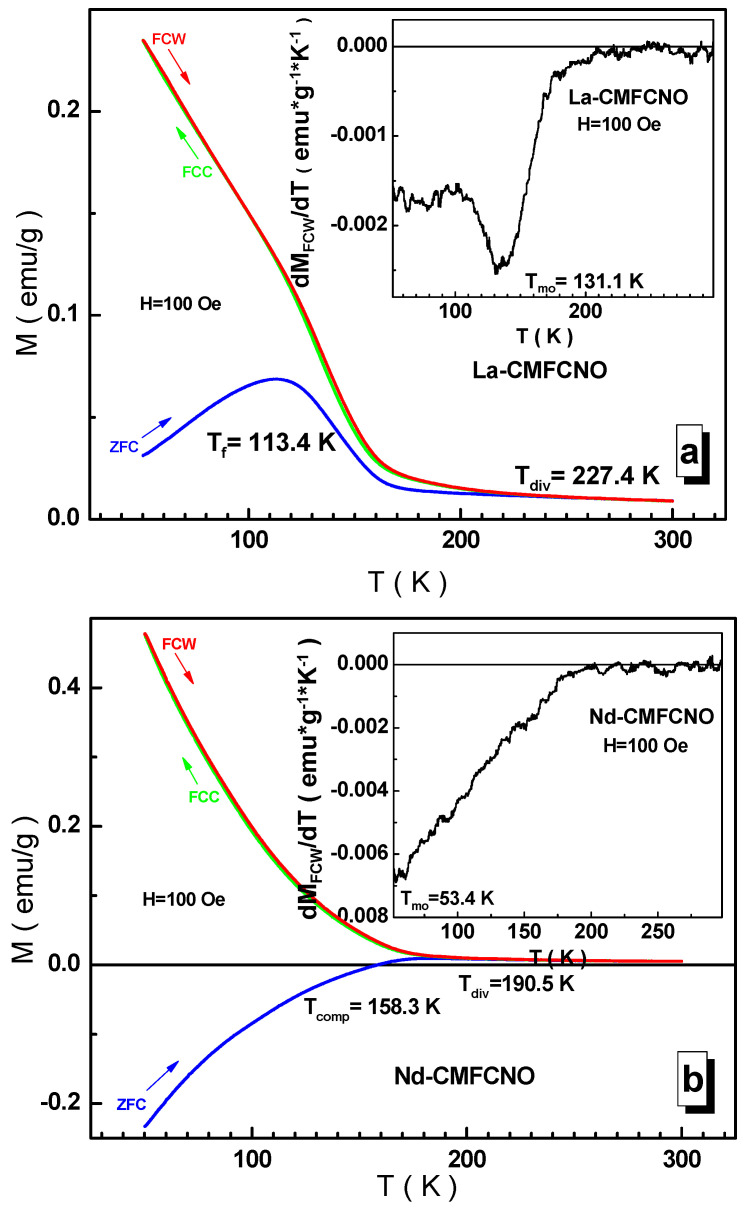
Temperature dependences of the specific magnetization in Zero-Field-Cooling and Field-Cooling regimes (FC-ZFC) for the La-CMFCNO (**a**), Nd-CMFCNO (**b**) and LN-CMFCNO (**c**) samples at 100 Oe. Insert demonstrates the temperature derivative of the corresponding FC curve.

**Figure 10 nanomaterials-11-01014-f010:**
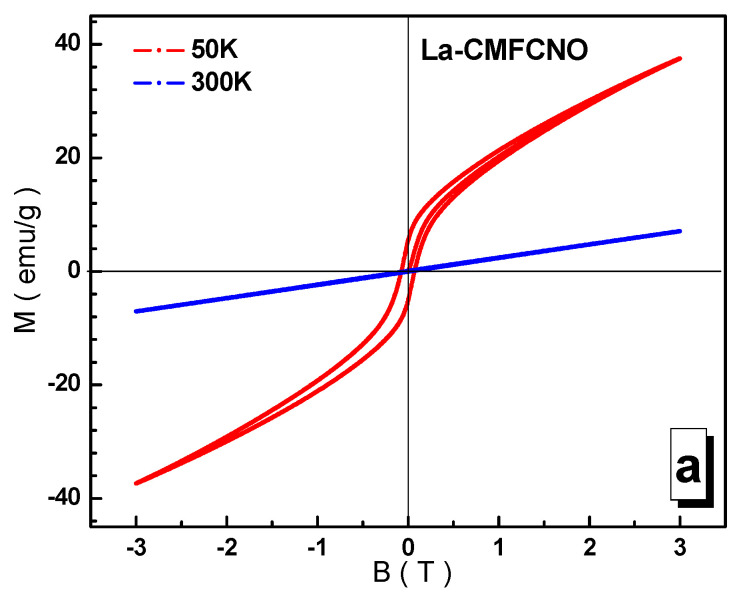
Field dependences of the specific magnetization for the La-CMFCNO (**a**), Nd-CMFCNO (**b**) and LN-CMFCNO (**c**) samples at 50 K (red line) and 300 K (blue line).

**Figure 11 nanomaterials-11-01014-f011:**
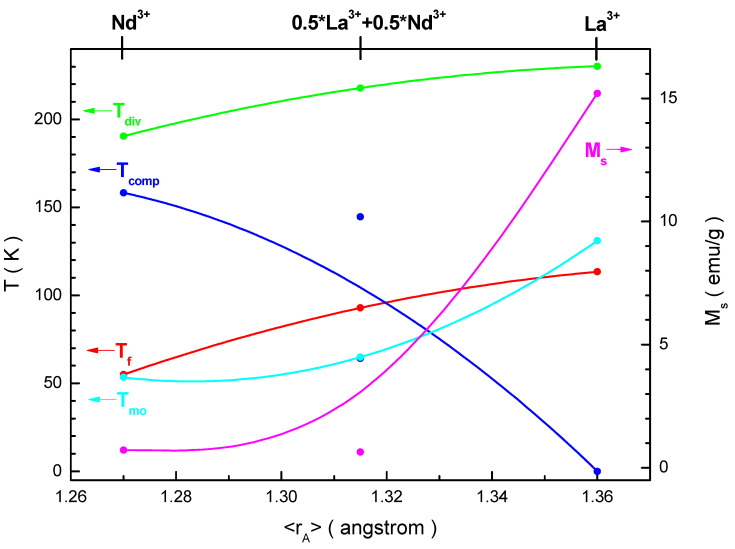
Dependences of the critical temperatures, such as the T_f_ freezing (red line), T_div_ divergence (green line), T_mo_ magnetic ordering (cyan line), T_comp_ compensation (blue line) temperatures (right axis) and M_s_ spontaneous magnetization (magenta line) (left axis) on the <r_A_> average ionic radius of the A-sublattice of perovskite structure for the investigated samples.

**Table 1 nanomaterials-11-01014-t001:** Correlation of the nominal chemical composition (real formula) and initial stoichiometry (calculated formula) of the La-CMFCNO, Nd-CMFCNO and LN-CMFCNO ceramic samples.

Formula	Cr	Mn	Fe	Co	Ni	La/Nd
**La-CMFCNO**						
**Calculated**	La(Cr_0.2_Mn_0.2_Fe_0.2_Co_0.2_Ni_0.2_)O_3_					
**at. %**	4.85 + 0.3	4.44 + 0.5	4.09 + 0.3	3.44 + 0.4	3.52 + 0.3	La-20.4 + 0.7
**Real**	La_1.04_(Cr_0.22_Mn_0.21_Fe_0.18_Co_0.17_N_i0.19_)O_3_					
**Nd-CMFCNO**						
**Calculated**	Nd(Cr_0.2_Mn_0.2_Fe_0.2_Co_0.2_Ni_0.2_)O_3_					
**at. %**	4.23 + 0.5	4.07 + 0.4	3.88 + 0.6	3.71 + 0.4	3.64 + 0.3	Nd-19.92 + 0.8
**Real**	Nd_1.01_(Cr_0.21_Mn_0.21_Fe_0.21_Co_0.19_Ni_0.18_)O_3_					
**LN-CMFCNO**						
**Calculated**	[La_0.5_Nd_0.5_](Cr_0.2_Mn_0.2_Fe_0.2_Co_0.2_Ni_0.2_)O_3_					
**at. %**	4.02 + 0.6	4.02 + 0.5	3.97 + 0.5	3.87 + 0.5	4.03 + 0.4	La-10.36 + 0.8Nd-9.56 + 0.8
**Real**	[La_0.53_Nd_0.48_](Cr_0.21_Mn_0.20_Fe_0.19_Co_0.19_Ni_0.2_)O_3_					

**Table 2 nanomaterials-11-01014-t002:** Lattice parameters, relevance factors and atomic coordination of the La-CMFCNO, Nd-CMFCNO and LN-CMFCNO ceramic samples.

La(Cr_0.2_Mn_0.2_Fe_0.2_Co_0.2_Ni_0.2_)O_3_—SG: R-3c									
Lattice parameters and relevance factors									
a		b		c	χ^2^		R_p_	R_wp_	R_exp_
5.4977 + 0.0009		5.4977 + 0.0009		13.3801 + 0.0029	2.89		6.94	9.69	8.94
Atomic coordination									
O1	0.4542		0.0000			0.2500			
La	0.0000		0.0000			0.2500			
Mn	0.0000		0.0000			0.0000			
Cr	0.0000		0.0000			0.0000			
Fe	0.0000		0.0000			0.0000			
Co	0.0000		0.0000			0.0000			
Ni	0.0000		0.0000			0.0000			
**Nd(Cr_0.2_Mn_0.2_Fe_0.2_Co_0.2_Ni_0.2_)O_3_—SG: Pnma**									
Lattice parameters and relevance factors									
a		b		c	χ^2^		R_p_	R_wp_	R_exp_
5.4739 + 0.0013		7.6694 + 0.0019		5.4076 + 0.0014	2.73		6.99	9.5	5.75
Atomic coordination									
O1	0.1934		0.0380			0.2900			
O2	0.5139		0.2500			0.5566			
Nd	0.4579		0.2500			0.0100			
Mn	0.0000		0.0000			0.0000			
Cr	0.0000		0.0000			0.0000			
Fe	0.0000		0.0000			0.0000			
Co	0.0000		0.0000			0.0000			
Ni	0.0000		0.0000			0.0000			
**[La_0.5_Nd_0.5_](Cr_0.2_Mn_0.2_Fe_0.2_Co_0.2_Ni_0.2_)O_3_—SG: Pnma**									
Lattice parameters and relevance factors									
a		b		c	χ^2^		R_p_	R_wp_	R_exp_
5.4712 + 0.0023		7.7072 + 0.003		5.4548 + 0.0023	2.18		8.03	11.1	11.76
Atomic coordination									
	x		y			z			
O1	0.2105		0.0375			0.2771			
O2	0.5056		0.2500			0.5774			
La	0.4679		0.2500			0.0060			
Nd	0.4679		0.2500			0.0060			
Mn	0.0000		0.0000			0.0000			
Cr	0.0000		0.0000			0.0000			
Fe	0.0000		0.0000			0.0000			
Co	0.0000		0.0000			0.0000			
Ni	0.0000		0.0000			0.0000			

**Table 3 nanomaterials-11-01014-t003:** Activation energies of SPH (Small Polaron Hopping) and VRH (Variable Range Hopping) mechanisms for La-CMFCNO.

Temperature Range	Activation Energy, E_A_ meV
$T<\frac{T_{D}}{4}$	46.7
$\frac{T_{D}}{4}<T<\frac{T_{D}}{2}$	156.5
$T>\frac{T_{D}}{2}$	122.8

## Data Availability

Not applicable.

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
