# Peer review of "Polysubstituted High-Entropy [LaNd](Cr0.2Mn0.2Fe0.2Co0.2Ni0.2)O3 Perovskites: Correlation of the Electrical and Magnetic Properties"

_nanomaterials, 2021, doi:10.3390/nano11041014_

Round 1

Reviewer 1 Report

To the Authors, 

I send my comments in the attached PDF file

regards

Author Response

Referee #1 Comments and Suggestions for Authors

The authors synthesized high-entropy phases with a perovskite-like crystal structure with the goal to achieve a single-phase composition. By solid state reaction they synthesize the La-, Nd- and La/Nd-based poly-substituted [LaNd](Cr0.2Mn0.2Fe0.2Co0.2Ni0.2)O 3 HEO’s named respectively La-CMFCNO, NdCMFCNO and LN-CMFCNO. After confirming via EDX measurements that the nominal compositions agree well with the initial calculated stoichiometries, they perform a series of characterizations: XRD analysis, to assess their crystal structures SEM microscopy , to investigate the microstructure and the particle size distribution Calculations of the configurational entropy Measurements of conductivity Measurements of magnetization.

I think that this is a solid experimental work. Moreover, the data analysis appears to be accurate., and the figures are well done and clear. Unfortunately, the manuscript is affected by serious problems in the presentation that strongly affect the readability and cannot be published in the present form.

Response: Dear Reviewer, thank you for your comment. Hope that revised version will be accepted in Nanomaterials.

Comment 1: The major problems to be solved are:

o The introduction should be improved. The authors offer a wide overview, but do not write why they chose to concentrate on La-CMFCNO, Nd-CMFCNO and LN-CMFCNO. – Thank you for this comment. We highlighted in Introduction our motivation of our choice:

The main idea of the choice of research objects is to establish a correlation between the ionic radius of the A-cation (La, Nd, and the La0.5Nd0.5 intermediate composition), the state of configurational entropy, and the features of the magnetic and transport properties. Thus, the La3+ ion is a diamagnetic ion and is characterized by the maximum ionic radius and leads to distortion of the lattice, which in turn leads to an increase in exchange interactions. The Nd3+ ion is a paramagnetic ion, and at low temperatures (below 50 K) magnetic ordering effects can be observed (the neodymium sublattice can be ordered antiparallel to the manganese sublattice, which leads to compensation effects). The aim of the study of the intermediate compound (La0.5Nd0.5 perovskite) is to evaluate the simultaneous influence of the diamagnetic lanthanum ion and the paramagnetic neodymium ion in the structure of high-entropy perovskite.

o The authors should limit the scattering of the discourse – Thank you for this comment. We did this.

o Many statements require a reference – Thank you for this comment. We added some references for statements.

o Minor problems: typos and acronyms that are not explained, etc. There are also some problems with English language, but they are of secondary importance here – Thank you for this comment. We revised this.

Response 1: Dear Reviewer, thank you for your comment.

Comment 2: Here is a list of observations, in order of appearance:

Line 103 “All the chemicals used were of analytical grade” the authors should be more specific and write the chemicals purities. – All the chemicals used were of analytical grade (99,999%).

Line 118-120 “The composition of the material was selected based on the need to maximize the configurational entropy of mixing of one of the cationic sublattices”. On which basis the configurational entropy of mixing is maximized by the chosen composition? - The text is supplemented with explanations of why the selected compositions should have the maximum configurational entropy of mixing.

Line 121-122: the configurational entropy of mixing for a cation lattice of five components is presented as equal to 1.609R. This is clearly derived from Boltzmann’s equation, but he authors should explain R and how they calculated it. [see below] - The text is supplemented with the formula on which the calculation was based (it was moved from clause 3.2 and corrected). The R value is deciphered in the text.

Line 137-142 [continued from above] the authors resume the entropy discourse with an assumption: they present the sublattice model and they adopt it. The authors should limit the scattering of the presentation. This text and the text at line 121-122 would fit more in paragraph 3.2; the discourse would be more readable. - The fragment in item 2 explains why we chose these particular quantitative compositions. The calculation is performed for the ideal case before the experiment. This applies to research methods. On the contrary, Section 3.2 discusses the actually obtained results and presents the results of calculating the configurational entropy of mixing, based on the actually obtained compositions of materials. This applies to the results of the study. Changes have been made to the text for a clearer distinction between the meanings of these two fragments. Added calculation results using actual data.

Line 214: “the sample consists of major particles[…]”. Do the authors mean “the sample consists mostly of particles […]”? – Yes. We do.

Line 215: The authors state that the main peak of Fig3c are particles of average size 4μ without explaining also the low-size tail. At line 215 they write that they are “surrounded by agglomerates of particles from a finer phase” without relating it to the tail. The authors should rewrite this sentence more clearly. – Thank you for this comment. We did this.

Line 245-278: this text is very confused. Some of the main problems are:

â–ª Line 250-255: Jonscher power law is not described properly. The dispersive component is written as and the authors do not write that “f” is the frequency (see Jonscher, Universal Relaxation Law, 1996).  - Thank you for finding this mistake,   f is defined.

â–ª Line 259: in “inverse functional dependence” I think that the authors mean “opposite functional dependence” - Yes, the referee is right

 â–ª “There was no any correlation between frequency and the value of the conductivity at fixed temperature for the Nd-CMFCNO (Fig. 4b)” apart of the typo “no any”, such a sentence is rather obscure and should not appear in any scientific manuscript. â–ª Line 269: “is standard situation for the semimetals (perovskites)” this statement requires a reference. – Thank you. The sentence is corrected

â–ª The text describes loosely the electric conductivity spectra of Fig4 as a function offrequency (from 100 to 107 Hz) and different temperatures (from 123 to 573 K). At line 272, after the description, the authors state that “in this work, we will completely concentrate on the investigation of the DC conductivity” and “The studies of the AC conductivity will be subject of the other paper”. This is rather confusing and it is not clear why the authors took the time to describe these spectra without explaining them. They should have avoided Fig4 and this description, and they should have shown directly the DC conductivities of Fig5 – Many thank for this comment. In some sense, the referee is right. However, the experimentally measured spectra (La-CMFCNO) is very unusual. It is impossible to explain them. The did not follow either universal Jonscher law or classical Drude model. For this reason, it is desirable that readers see these spectra which could motivate them to try to explain them.

â–ª more typos, not listed here.

 Line 287 “Temperature dependences of the DC for the La-CMFCNO[…]” the word conductivity is missing. – Thank you. It is corrected

Line 309 “[…]the DC conductivity obey the Mott-VRH law” requires a reference

Response 2: Dear Reviewer, thank you for your comments. We revised all features which were mentioned by you.

Comment 3: Figures

Fig 6and Fig 9: the axis label and numbers of the insets will appear too small in a paper format.

Response 3: Dear Reviewer, thank you for your comment. We revised this.

Comment 4: Minor problems

Line 51 “Transition ions” “Transition metal ions” – Thank you for this comment. We revised this

Line 62 “substation” for “substitution” – Thank you for this comment. We revised this

Line 70 “valent angles”; “bond angles” should be more appropriate – Thank you for this comment. We revised this

Line 127: 1300 is written without units – Thank you for this comment. We revised this

Line 129: in “Cu-Ka” Ka is obviously Kα – Thank you for this comment. We revised this

Line 132: the acronym EDX is not explained – Thank you for this comment. We revised this

Line 180 “good fitting appropriate values” something is clearly missing – Thank you for this comment. We revised this

Line 344: The acronyms FC and ZFC are not explained – Thank you for this comment. We revised this

Line 371: “see inset in Fig. 8” but it is the inset in Fig. 9 – Thank you for this comment. We revised this

Line 194 in “particle” the “s” is missing – Thank you for this comment. We revised this

Response 4: Dear Reviewer, thank you for your comment. We revised all details mentioned above.

Many thanks for your comments. All comments were useful for us and we hope it helped us do our manuscript better.

Reviewer 2 Report

In this paper, Zhivulin et al report preparation, structure, electrical and magnetic properties of the La-, Nd- and La/Nd-based polysubstituted high-entropy oxides (HEO’s)  produced by solid  state reactions. The compositions of these compounds were found be nearly consistent with the initial calculated stoichiometry. The results showed that the structure was a single phase.  The temperature and frequency dependent electrical properties were also investigated. The conductivity  mechanism was attributed to variable range hopping and  small polaron hopping. The observed frustrated  state of the spin subsystem was  due to the increase of the entropy state. The work is comprehensive and interesting. It merits the publication in Nanomaterials, providing the follows are addressed.

(1) In the abstract, "It was calculated the average value of the configurational entropy for all samples."  This sentence is not clear.

(2) In Line 127, 1300 should be 1300oC.

(3) In Line 137, 473oK should be "473K". Please unify the temperature unit to be oC or K in the whole text.

(4) In figure 1 of the XRD patterns, please mark the peaks orientation.

(5) For the EDX, did you detect the nitrogen content? since the annealing experiments were done in air.

(6) In fig. 4, for the conductivity reduction decrease at high frequency, the reason might be the traps from grain boundaries or point defects, which can not follow the high frequency signal. Please comment this point.

(7) Can you comment the potential applications of  these oxide in MEMS magnetic sensing applications near room temperatures  (Refs.  doi.10.1080/26941112.2021.1877019 and doi:10.3390/mi11050523). This may enhance the impact your work.

Author Response

Referee #2 Comments and Suggestions for Authors

In this paper, Zhivulin et al report preparation, structure, electrical and magnetic properties of the La-, Nd- and La/Nd-based polysubstituted high-entropy oxides (HEO’s)  produced by solid  state reactions. The compositions of these compounds were found be nearly consistent with the initial calculated stoichiometry. The results showed that the structure was a single phase.  The temperature and frequency dependent electrical properties were also investigated. The conductivity  mechanism was attributed to variable range hopping and  small polaron hopping. The observed frustrated  state of the spin subsystem was  due to the increase of the entropy state. The work is comprehensive and interesting. It merits the publication in Nanomaterials, providing the follows are addressed.

Comment (1): In the abstract, "It was calculated the average value of the configurational entropy for all samples."  This sentence is not clear.

Response 1: We revised abstract. Now it seems like: It was calculated the values of the configurational entropy of mixing for each sample.

Comment (2): In Line 127, 1300 should be 1300oC.

Response 2: Thank you for your attention. We revised this.

Comment (3): In Line 137, 473oK should be "473K". Please unify the temperature unit to be oC or K in the whole text.

Response 3: Thank you for your comment. We revised "473K". But we used C for synthesis processes and K for measurement procedure.

Comment (4): In figure 1 of the XRD patterns, please mark the peaks orientation.

Response 4: Thank you for your comment. In investigated samples we do not observe any crystallite orientation (texture) that is why we do not highlighted any preferable orientation. Hope on your understanding.

Comment (5): For the EDX, did you detect the nitrogen content? since the annealing experiments were done in air.

Response 5: Thank you for this comment. We did not record the nitrogen content in the EDX spectra.

Comment (6): In fig. 4, for the conductivity reduction decrease at high frequency, the reason might be the traps from grain boundaries or point defects, which can not follow the high frequency signal. Please comment this point.

Response 6: The reason for this phenomenon is really unclear. In most articles it was tried to explained by the classical Drude model, but it is not our case because we have hopping mechanism of conductivity  but not drift- diffusion mechanism.

From the other side, in any case, one should observe interfacial polarization, which could also include contribution due to trapping. But even in this case, the real part of the conductivity spectrum should folloe the universal Jonscher law. However, we see opposite behavior and currently  we are unable to explain it.

Comment (7): Can you comment the potential applications of  these oxide in MEMS magnetic sensing applications near room temperatures  (Refs.  doi.10.1080/26941112.2021.1877019 and doi:10.3390/mi11050523). This may enhance the impact your work.

Response 7: Thank you for this useful information. We discussed perspectives of the practical applications. Please see in text and ref [61, 62].

Round 2

Reviewer 2 Report

Now this paper can be accepted.